# Cellular Models and Assays to Study NLRP3 Inflammasome Biology

**DOI:** 10.3390/ijms21124294

**Published:** 2020-06-16

**Authors:** Giovanni Zito, Marco Buscetta, Maura Cimino, Paola Dino, Fabio Bucchieri, Chiara Cipollina

**Affiliations:** 1Fondazione Ri.MED, via Bandiera 11, 90133 Palermo, Italy; gzito@fondazionerimed.com (G.Z.); mbuscetta@fondazionerimed.com (M.B.); mcimino@fondazionerimed.com (M.C.); 2Dipartimento di Biomedicina Sperimentale, Neuroscenze e Diagnostica Avanzata (Bi.N.D.), University of Palermo, via del Vespro 129, 90127 Palermo, Italy; paola.dino@unipa.it (P.D.); fabio.bucchieri@unipa.it (F.B.); 3Istituto per la Ricerca e l’Innovazione Biomedica-Consiglio Nazionale delle Ricerche, via Ugo la Malfa 153, 90146 Palermo, Italy

**Keywords:** NLRP3, inflammasome, NLRP3 inhibitors, cell models, biochemical assays, biophysical assays, read-outs

## Abstract

The NLRP3 inflammasome is a multi-protein complex that initiates innate immunity responses when exposed to a wide range of stimuli, including pathogen-associated molecular patterns (PAMPs) and danger-associated molecular patterns (DAMPs). Inflammasome activation leads to the release of the pro-inflammatory cytokines interleukin (IL)-1β and IL-18 and to pyroptotic cell death. Over-activation of NLRP3 inflammasome has been associated with several chronic inflammatory diseases. A deep knowledge of NLRP3 inflammasome biology is required to better exploit its potential as therapeutic target and for the development of new selective drugs. To this purpose, in the past few years, several tools have been developed for the biological characterization of the multimeric inflammasome complex, the identification of the upstream signaling cascade leading to inflammasome activation, and the downstream effects triggered by NLRP3 activation. In this review, we will report cellular models and cellular, biochemical, and biophysical assays that are currently available for studying inflammasome biology. A special focus will be on those models/assays that have been used to identify NLRP3 inhibitors and their mechanism of action.

## 1. Introduction

Innate immunity represents the first line of defense against invading pathogens or endogenous stress signals. Innate immune responses are mediated by a series of biological processes that have the common aim of restoring tissue homeostasis. The inflammatory cascade is triggered by the recognition of pathogen-associated molecular pattern (PAMP) and danger-associated molecular pattern (DAMP) by pattern recognition receptors (PRRs) that are mainly expressed by immune cells, such as macrophages. Among PRR, the nucleotide-binding domain and leucine-rich repeat containing receptors (NLRs) family is able to recognize cytosolic DAMPs/PAMPs. NLRs are expressed in the cytosol of myeloid-derived immune cells as well as in other cell types, such as epithelial and endothelial cells. Among NLRs, NLRP3 is one of the most studied and represents an attractive therapeutic target for several chronic diseases. Upon activation, NLRP3 assembles into a multimeric inflammasome complex comprising a core unit containing the adaptor apoptosis-associated speck-like protein containing a CARD (ASC) and the effector pro-caspase-1. In specific cases, NIMA-related kinase 7 (NEK7) binding to NLRP3 appears to be required for NLRP3 activation [1]. Following inflammasome assembly, autocatalytic activation of caspase-1 takes place, triggering the cleavage and release of the pro-inflammatory cytokines IL-1β and IL-18, and the processing of gasdermin-D (GSDMD), which leads to a form of programmed inflammatory cell death called pyroptosis.

Gain of function mutations of NLRP3 genes cause cryopyrin associated periodic syndromes (CAPS) [2]. Over-activation of NLRP3 has been associated with many chronic inflammatory diseases such as Alzheimer’s disease [3], Parkinson’s disease [4], multiple sclerosis [5], metabolic disease and type 2 diabetes mellitus (T2D), atherosclerosis [6], gout [7], osteoarthritis [8], and rheumatoid arthritis [6,7,9]. These evidences, combined with genetic proofs that knocking out NLRP3 restores healthy phenotype in several disease models and the finding that downregulation of NLRP3 has minor impact on host defense mechanisms [10] make NLRP3 an attractive therapeutic target. Although numerous factors of NLRP3 biology have been extensively described, many aspects remain subject of debate. This review aims to summarize recent findings in NLRP3 inflammasome biology with a special focus on the tools (cellular models and assays) developed so far to study inflammasome activation and the action of small molecule inhibitors.

### 1.1. Mechanisms of NLRP3 Activation

Three pathways of NLRP3 inflammasome activation have so far been described: canonical, non-canonical and alternative pathway (Figure 1).

Canonical activation is the classical two-step model where two signals are required for optimal activation of the NLRP3 inflammasome. Signal 1, or priming, requires binding of toll-like receptors (TLRs) with pathogen-associated molecular patterns (PAMPs) such as lipopolysaccharide (LPS). Signal 1 induces the transcriptional up-regulation of NLRP3, pro-IL-1β, and pro-IL-18 via Nuclear Factor-kB (NF-kB) activation [11,12]. Growing evidence indicates that Signal 1 promotes more than transcriptional up-regulation, as it induces a number of post-translational modifications (PTMs) that allow NLRP3 to switch into its active conformation [13]. Signal 2 is triggered by diverse stimuli including PAMPs, DAMPs, and particulate matter which NLRP3 “senses” via yet undefined mechanisms. Signal 2 leads to the formation of the active inflammasome complex and the auto-proteolytic cleavage of caspase-1. A typical feature of the NLRP3 inflammasome is the ability to respond to a wide range of signals such as extracellular adenosine triphosphate (ATP), microbial toxins, crystals, particulate matter, and viral proteins. The exact molecular mechanism that triggers NLRP3 activation in response to such a diverse set of signals is still under investigation. Many NLRP3 activators induce K^+^ efflux. The consequent drop in intracellular K^+^ has been at first identified as a common trigger for NLRP3 inflammasome activation [14,15]. However, growing evidence has shown that, along with K^+^ efflux, other mechanisms may contribute to NLRP3 activation such as Cl^−^ efflux, Ca^2+^ signaling, reactive oxygen species (ROS) mitochondrial dysfunction, and lysosomal rupture [16,17]. Given the diversity of these signals, it is likely that NLRP3 “senses” a common pathway induced in the cytosolic environment by intracellular processes rather than directly interacting with all these molecules. In all cases, optimal activation of NLRP3 inflammasome requires multiple PTMs such as de-sumoylation [18], de-ubiquitination [19,20], phosphorylation [21,22], de-phosphorylation, acetylation [23], and alternative splicing [17,24]. Different PTMs have been accurately described in the literature and are reviewed in [13,17].

Non-canonical activation is triggered by caspase-4 in humans and caspase-11 in mice and occurs in response to intracellular infection by Gram-Negative bacteria (e.g., *Escherichia coli)* [25]. It has been reported that caspase-11 [26] and caspase-4 [27] are activated by intracellular LPS through direct binding of LPS with their CARD domain. Furthermore, it has been recently shown that other components of Gram-negative bacteria, as well as exogenous drugs, can activate caspase-4 and caspase-11 [28,29]. Several works, nicely summarized in Yi, 2020 [30], have shown that the non-canonical pathway in mice cooperates with the NLPR3 inflammasome in order to provide a robust inflammatory response. In fact, caspase-11/4 activation mediated by iLPS can promote K^+^ efflux, either by GSDMD cleavage and consequent pyroptosis or by currently unknown mechanisms leading to membrane rupture. As a consequence of K^+^ efflux, NLRP3 inflammasome becomes activated [27,31].

Alternative inflammasome activation is a new species-specific NLRP3 inflammasome pathway that was first reported in 2016. It exists in human and porcine peripheral blood mononuclear cells (PBMCs), but it is absent in murine ones [32]. In this pathway, LPS per se is sufficient to induce activation of the NLRP3 inflammasome with consequent activation of caspase-1 and IL-1β processing and secretion. Inflammasome assembly occurs upon activation of the TLR4 by LPS triggering the TIR-domain-containing adapter-inducing interferon-β (TRIF)—receptor-interacting serine/threonine-protein kinase 1 (RIPK1)—Fas-associated protein with death domain (FADD) caspase-8 signaling cascade, which in turns leads to the activation of the NLRP3 inflammasome. This pathway is not dependent on K^+^ efflux. No pyroptosis occurs, thus IL-1β is released gradually, as opposed to the all-or-nothing response of the canonical activation [32].

### 1.2. Role of Domains

NLRP3 has a N-terminal effector pyrin domain (PYD), which interacts with ASC via PYD–PYD interaction, a central NACHT domain carrying the ATPase activity, and a C-terminal leucine-rich repeats (LRR) domain.

The NLRP3-PYD domain recruits ASC via PYD–PYD interaction and it is therefore required for the formation of the active inflammasome complex [33]. It consists of a six-helical bundle structural fold containing several conserved residues as compared to other PYD domains interacting with ASC and with a possible homodimeric interface [34]. Due to its relevance for the activation of the NLRP3 inflammasome, the PYD domain represents an attractive target for the development of NLRP3 inhibitors, as recently reported [35].

The central NACHT domain provides the ATPase activity that is required for NLRP3 activation and inflammasome formation. The NACHT domain contains a walker A motif responsible for ATP binding and a Walker B motif that is necessary for ATPase activity [36]. An intact and functional NACHT domain is required for interaction with ASC, activation of caspase-1, and IL-1β release in THP-1 cells [37]. Of note, mutations of the NACHT domain are associated with spontaneous NLRP3 activation observed in CAPS [37]. Finally, it has recently been reported that the NACHT domain is involved in NLPR3 activation in response to viral infection through its binding with viral DexD/H-box helicase (DHX) proteins [38]. Current knowledge supports the hypothesis that the NACHT domain is a primary druggable site for the development of selective inhibitors of NLRP3.

The LRR domain is evolutionarily conserved in several different proteins that serve as pattern recognition receptors and typically harbors the sensing domain. Structurally, the LRR domain is a large β-helical array with horseshoe or arc shape [36,39]. The role of NLRP3-LRR is still under investigation. NLRP3-LRR has been proposed to be involved in auto-regulation, protein–protein interaction, and signal sensing. LRR appears to be dispensable for canonical NLRP3 activation. In fact, a truncated form of NLRP3 (residues 1–686, lacking the LRR domain) can be fully activated by the canonical pathway, indicating that LRR is not necessary for sensing and assembling of the inflammasome [40]. Nonetheless, LRR domain is involved in the recognition of microbial ligands through direct binding. For example, it has been reported that viral 3D RNA polymerase of Enterovirus 71 (EV71) associates with LRR domain, forming a “3D-NLRP3-ASC” ring-like structure [41]. Very recently, it has been shown that SARS-CoV open reading frame-8b (ORF-8b) binds the LRR domain and localizes with NLRP3 and ASC in cytosolic dot-like structures, suggesting that this interaction is functionally relevant for IL-1β release in response to the virus [42]. Finally, a possible inflammasome-independent function for the LRR domain has been described referring to the binding of LRR to the transcription factor IRF4, thus promoting the activity of CD4^+^ T_H_2 cells via IL-4 transcription [43].

### 1.3. Inhibition of NLRP3 for the Treatment of Inflammatory Diseases

Growing pre-clinical evidence indicates that inhibition of NLRP3 displays therapeutics benefits in several disease models while causing minimal impairment of host immune responses [10]. Along the same line, a number of clinical studies have shown that agents blocking IL-1β are effective for the treatment of several conditions including rheumatic diseases and autoinflammatory syndromes, and decrease the incidence of atherosclerotic disease in at risk patients with an inflammatory signature [44]. However, blockade of IL-1β signaling is associated with an increase of infections and sepsis [45]. Therefore, therapies selectively targeting the NLRP3 inflammasome, rather than downstream cytokine effectors, would display improved safety by preserving host immune defenses. Furthermore, they would provide increased therapeutic potency due to the simultaneous inhibition of IL-1β, IL-18, and pyroptosis. As a consequence, NLRP3 appears to be an ideal target for drug discovery. In recent years, a wide range of molecules has been developed as NLRP3 inhibitors (reviewed in [46]). To date, MCC950 is the most potent and selective inhibitor of NLRP3. MCC950 directly binds to NLRP3 NACHT domain affecting the ATPase function, leading to an inactive NLRP3 conformation [47,48,49]. Other NLRP3 inhibitors have been developed that directly target the NACHT domain, including CY-09, Oridonin, Tranilast, OLT1177 and MNS [50,51,52,53]. Other compounds inhibit the activation of the NLRP3 inflammasome via indirect mechanisms, either by blocking other components of the inflammasome complex or by inhibiting the signaling cascade leading to NLRP3 activation [54,55,56,57,58]. Very recently, it has been reported that targeting the PYD domain may represent an effective strategy for NLRP3 inhibition [35]. The efficacy of selective NLRP3 inhibitors for the treatment of NLRP3-related inflammatory diseases, such as neurodegenerative diseases, gouty arthritis, CAPS and metabolic disease, has been widely demonstrated in several preclinical models and ex-vivo systems [17,50,51,59,60]. However, despite great effort, none of the newly developed inhibitors of NLRP3 have so far been approved by the Food and Drug Administration (FDA). Therefore, current research should aim either at the improvement of the pharmacokinetic properties of the available molecules or at the discovery of novel classes of NLRP3 inhibitors, to reach a selective, potent and cost-effective remedy for a wide range of human diseases.

## 2. Cell Models to Study NLRP3 Inflammasome Biology

In this section, we will highlight the main cell models that are currently in place for inflammasome studies as summarized in Table 1. In addition, we will bring up new cell models that are currently under development and that may be useful for the creation of novel ex-vivo models of NLRP3-driven diseases. Finally, we will discuss about potential pitfalls and challenges when it comes to the choice of the proper model. Although inflammasomes are expressed both in myeloid and non-myeloid cell types, myeloid-derived cells and in particular macrophages are the cell type expressing the highest levels of NLRP3 and releasing the highest amount of cytokines. Therefore, these are normally used as model system for the study of inflammasome activation.

### 2.1. Murine Cell Models

#### 2.1.1. Primary Murine Bone-Marrow-Derived Macrophages (BMDMs)

Primary murine bone-marrow-derived macrophages (BMDMs) have been a very useful tool for the study of NLRP3 inflammasome biology. Established protocols exist to obtain BMDMs from femurs and tibia of C57BL/6 mice. An advantage of using BMDMs is represented by the fact that knock-out BMDMs can be obtained from mice genetically deficient in inflammasome components such as Caspase 1^-/-^, ASC^-/-^, and NLRP3^-/-^ [14,26,61]. For this reason, BMDMs have been of primary importance for the elucidation of the molecular mechanisms underlying NLRP3 inflammasome activation [1,14,15,18,19,22,24,26,54]. For example, BMDMs have been used to define the precise signals that activate canonical NLRP3 inflammasome pathway, including K^+^ efflux and Ca^2+^ mobilization [15,62]. Furthermore, BMDMs were used to evaluate NLRP3 activation in non-canonical inflammasome pathway to demonstrate that caspase-11 plays a role in modulation of K^+^ efflux and to prove that GSDMD is essential for caspase-11-dependent pyroptosis and for IL-1β maturation [26,63,64].

Given the high costs of animal housing and breeding, and considering that it is always better to work with fresh primary cells, in the past few years, several protocols for BMDM immortalization have been developed [65,66,67]. Briefly, the approach uses the Cre-J2 retroviral method of infection using a J2 retrovirus carrying v-raf and v-myc oncogenes. This method, developed in the 1980s and recently improved, allows the generation of iBMDMs phenotypically comparable to their primary counterparts, displaying many of the trademark functions of macrophages [68]. iBMDMs have been widely used to study NLRP3 PTMs and identify NLRP3 inhibitors [1,18,21,22,40,49,52,69,70].

#### 2.1.2. Murine Macrophage Cell Lines

Currently, two mouse cell lines are mostly used for the study of NLRP3 inflammasome biology—RAW264.7 and J774A.1. They are both macrophage-like adherent cell lines, very easy to grow and manipulate, and commonly used to study the innate immune responses. Despite their use in inflammasome research, they differ for a very important molecular aspect: RAW264.7 cells do not express the ASC protein, while J774A.1 cells do express it [61,71]. As a consequence, RAW264.7 cells can be used to study the transcriptional events that regulate priming, but not the downstream cascade, as they cannot activate the NLRP3 inflammasome complex. In addition, for this specific feature, the RAW264.7 cell line has been used to identify a novel mechanism of NLRP3-independent bacterial killing mediated by K^+^ efflux [72]. Interestingly, stable transfection of RAW264.7 cells with plasmids containing the full length sequence for ASC can restore the whole NLRP3 inflammasome machinery [73].

J774A.1 is a macrophage-like cell line able to form a complete functional NLRP3 inflammasome system [71]. These cells represent a reliable model to study inflammasome biology as they are easy to manipulate and grow [69,70,74]. For example, J774A.1 cells have been used by Yaron and collaborators to demonstrate that K^+^ efflux is upstream of Ca^2+^ influx in the production of mtROS, thus beginning the cascade of events leading to NLRP3 inflammasome activation [75]. J774A.1 cells have been widely used to test the efficacy of potentially novel inhibitors of NLRP3 [53,76,77,78]. For instance, Hu et al. nicely demonstrated that the antimicrobial cathelicidin peptide LL-37 inhibits LPS/ATP-mediated pyroptosis, thus providing new insights into modulation of sepsis [79].

### 2.2. Human Cell Models

While the murine cell models set the ground primarily for the characterization of the molecular mechanisms regulating NLPR3 inflammasome activation, the use of human models has been necessary to define the mechanisms of NLRP3-driven human diseases.

#### 2.2.1. Human Monocyte-Derived Macrophages (hMDMs)

Human monocytes can be obtained from freshly isolated PBMCs according to established protocols [12,80] and differentiated into human monocyte-derived macrophages (hMDMs). To this purpose, different approaches have been developed where different stimuli can be used in order to obtain a specific macrophage polarization [12,81,82]. For example, M-CSF-differentiated hMDMs treated with LPS/IFNγ or IL-4, become polarized toward the M1 or M2 phenotype, respectively [83,84]. The high plasticity of this cellular model can be used to recapitulate the different aspects of the immune response in vitro. hMDMs were used primarily to study NLRP3 canonical activation and inhibition [11,22,24,48,52,54] and became crucial for the identification of the non-canonical pathway of NLRP3 inflammasome activation mediated by caspase-4 [85]. hMDMs can be cultured for two to three weeks in vitro for experimental purposes. It is always recommended to work with fresh hMDMs. Nevertheless, they can be frozen in specific freezing culture for long-term storage.

#### 2.2.2. Human Monocyte/Macrophage Cell Lines

Considering the difficulties to obtain primary monocytes, given the high variability among donors and the fact that they are not amenable to genetic manipulation, the use of hMDMs has been limited in time. In the last few decades, a variety of human cell lines have been tested for their potential capability to activate NLRP3 inflammasome. Among others, the THP-1 and the U937 cell lines have been mostly used. The THP-1 monocyte-like cell line, derived from acute monocytic leukemia, has been extensively used in the field, despite its tumoral derivation and consequent genomic instability [86]. This cell line can be differentiated into macrophages by treatment with phorbol-12-myristate-13-acetate (PMA) [87,88]. Differentiated THP-1 cells display several features of primary hMDMs, as shown by macrophage marker expression, morphology, phagocytic activity, and cytokine release. When used for the study of inflammasome activation, THP-1 cells are usually differentiated with PMA into macrophages for a time ranging of 24–72h, primed with LPS, and then subjected to different second signals necessary for NLRP3 inflammasome activation [22,37,42,52]. For instance, Petrilli and collaborators in 2007 treated THP-1-derived macrophages with ionophores such as nigericin, gramicidin and valinomycin to demonstrate the key role of K^+^ efflux in triggering NLRP3 inflammasome activation [89]. THP-1 cells were also used to demonstrate that DAMPs and exogenous signals, including monosodium urate (MSU) crystals, asbestos, silica, and mitochondrial ROS, activate the NLRP3 inflammasome [90,91,92]. Furthermore, Iyer et al. used THP-1 cells to demonstrate that the treatment with the antibiotic Linezolid led to mitochondrial disruption, cardiolipin release and NLRP3 inflammasome activation in a ROS-independent fashion [93]. THP-1 cells can also be easily manipulated in vitro. Stable THP-1 knock-out cell lines targeting specific genes of the NLRP3 inflammasome cascade have been developed [27,94]. This has been instrumental for demonstrating that not only non-canonical NLRP3 inflammasome activation is mediated by caspase-4-mediated LPS intracellular sensing, but also that caspase-1 and -4 cleaves GSDMD, thus leading to pore formation and pyroptosis.

The U937 cell line is a pro-monocytic myeloid leukemia cell line that, similarly to THP-1 cells, has been used for inflammasome studies. U937 cells can be differentiated into macrophages with PMA, and inflammasome activation can be achieved by LPS stimulation followed by different second signals. U937-derived macrophages display similarities with hMDMs; thus, they have been used to study mechanisms of inflammasome activation and for the identification of novel NLRP3 inhibitors [22,53,95].

#### 2.2.3. BlaER1 Human Cell Model

THP-1 cells do not fully recapitulate the behavior of primary human monocytes, as they totally or partially lack several signaling cascades that are present in primary immune cells and are characterized by karyotypic abnormalities [86,96]. Thus, to fill the gap between cell lines and primary human myeloid cells, a new human cell model has been established [32,97]. This human cell model, called BlaER1, employed the stable expression of C/EBPα transcription factor in immortalized immune B cells [97,98]. Activation of C/EBPα induces trans-differentiation, causing BlaER1 cells to switch from their proliferative B-cell stage to a post-mitotic, monocytic status, in which they become moderately adherent, highly phagocytic, and competent for multiple innate immune signaling pathways. BlaER1 cells have been instrumental for the discovery of the alternative pathway of NLRP3 inflammasome activation mediated by the TLR4/TRIF/caspase-8 axis.

#### 2.2.4. HEK293T Cell Line

A very useful tool for in vitro studies of the molecular mechanisms of inflammasome activation is represented by the reconstitution of the NLRP3 inflammasome into the HEK293T human cell line. This cell line can be easily transfected, as it is commonly used for protein expression and production of recombinant retro/lentiviruses. Several research groups have used HEK293T cells to study NLRP3 inflammasome biology [37,40,42,48,49,52]. These cells do not express any of the inflammasome-related proteins, so it is necessary to transfect them with specific plasmids or retroviral constructs carrying the gene of interest in order to express the proteins to study. HEK293T cells have been used to reconstitute the entire NLRP3 mouse inflammasome system [99,100]. Different studies used HEK293T cells to identify NLRP3 post-translational modifications occurring during inflammasome activation and to study protein–protein interaction by evaluating co-localization or by performing co-immunoprecipitation assays [52,59,101]. Among others, Song et al. used transfected HEK293T cells to demonstrate that NLRP3 phosphorylation mediated by JNK1 is an essential priming event for inflammasome activation [22]. Wang et al. recently utilized HEK293T cells to prove that the stimulator of interferon genes (STING) binds to NLRP3 thus mediating its localization into the ER and determining its de-ubiquitination required for inflammasome activation [102]. Finally, Mao et al. used HEK293T cells to demonstrate that Bruton tyrosine kinase (BTK) binds to NLRP3 to regulate its activation, therefore suggesting that BTK deficiency is associated with several inflammatory NLRP3-mediated diseases [103].

#### 2.2.5. Induced Pluripotent Stem-Cells-Derived Macrophages (iPS-DM)

Despite the advances in cell models for the study of NLRP3 inflammasome biology, we still lack the “perfect” system able to recapitulate the features of primary macrophages while at the same time being able to replicate and being amenable to genetic manipulation. For example, with the tools we have in place, it is difficult to genetically manipulate primary hMDMs and, on the other side, the use of THP-1 cell line is limited, among others, by the fact that they are karyotypically abnormal [86,104]. For these reasons, several research groups have begun to establish macrophages from induced-pluripotent stem cells (iPS). Currently, different protocols have been tested to efficiently reprogram iPS into mature macrophages, and some of them nicely demonstrated how the iPS-DM showed similarities with hMDMs, including morphology, expression of surface markers, transcriptional and cytokine release profiles, and functional abilities, such as phagocytosis [105,106,107,108,109,110]. These studies opened the possibility of using iPS-DMs derived from healthy subjects as well as from patients for drug discovery purposes. For instance, in 2012, Tanaka et al. obtained iPS-DMs from patients affected by chronic infantile neurologic cutaneous and articular syndrome (CINCA), an IL-1β-driven inflammatory disease caused mainly by NLRP3 mutations leading to its constitutive activation. This study clearly showed the impact of NLRP3 mutation on the development of the pathology, and began to define new potential therapeutic approaches for this type of disease [111].

Despite the attempts and the different protocols in place to obtain iPS-DMs, few limitations need to be considered, including the differentiation efficiency (still far from acceptable) and the choice of the target genes to study (their expression or behavior have to be similar to hMDMs).

### 2.3. The Importance of Choosing the Right Model

Considering the abundance of cellular models available and their diversity (Table 1), the choice of the right tool becomes extremely important. As mentioned, mouse models represent a very useful tool to understand inflammasome biology, and murine cell lines are diverse regarding the expression of NLRP3 inflammasome components. Therefore, it is very important to use the right model according to the questions to address. Even when human cell models are used, the choice has to be well thought as for example alternative NLRP3 activation up to now has been reported only in BlaER1 cells. Finally, while some mouse models recapitulate key features of NLRP3-related human disease (i.e., CAPS), it is always recommended, when possible, to compare the data obtained with human in vitro/ex-vivo models or, even better, with samples derived from patients. In that sense, the establishment of iPS-DMs models will be extremely useful for in vitro studies of NLRP3-related auto-inflammatory diseases.

## 3. Cellular, Biochemical, and Biophysical Assays to Evaluate Activation and Function of the NLRP3 Inflammasome

Activation of the NLRP3 inflammasome is a complex event regulated at multiple levels. It requires the formation of a multimeric protein complex and leads to a cascade of events that can be monitored by evaluating multiple read-out. Over the years, several assays have been developed that allowed to study NLRP3 inflammasome biology (Table 2). In this section, we extensively review the different types of assays that are currently used. Whenever cell-based assays are in place, independent of whether murine or human macrophages, the canonical two-step model of activation is normally applied to activate the NLRP3 inflammasome. Briefly, cells are stimulated with LPS (usually for 3–5 h) to induce the priming followed by treatment with nigericin or extracellular ATP (for 1–2 h) to license the active NLRP3 inflammasome complex. Of note, if the experiments are aimed to assess the actions of selective NLRP3 inhibitors, these are generally added after the priming step and before the second stimulation with ATP or nigericin.

### 3.1. Immuno-Based Assays

Immune assays are based on the utilization of antibodies that specifically recognize and bind to a given protein of interest. Recognition of a target protein by a specific antibody is exploited in large number of assays; Enzyme-Linked Immunosorbent Assay (ELISA) is one of the most widely used assays that allows the user to monitor soluble proteins and cytokine release in cell supernatants. When applied to IL-1β and IL-18 quantification, this approach involves the use of antibodies that recognize the mature form of the cytokines. ELISA assay provides quantitative information on the concentration of cytokine present in cell supernatants. However, it is a long and multi-step procedure. Recently, different homogeneous assays have been developed for quantitative detection of soluble proteins, including IL-1β and IL-18, in a very rapid and sensitive procedure amenable to miniaturization [112,113]. They are based on time-resolved fluorescence resonance energy transfer (TR-FRET), as is the case of HTRF and Lance technologies (provided by PerkinElmer), and on the more sensitive bead-based luminescent amplification assay, as is the case of AlphaLISA (provided by PerkinElmer) [114,115]. Western blot on supernatant precipitates is the preferred approach to evaluate the processing of pro-IL-1β, pro-IL-18, GSDMD, and caspases. In this respect, the choice of antibodies will determine which one among the pro-form and the cleaved forms can be observed. Despite providing unique information regarding the extent of protein cleavage and the different subunits generated upon processing, this techniques is qualitative, as opposed to ELISA, TR-FRET-, and AlphaLISA-based assays [14,116,117]. Detection of ASC oligomers is very often used to assess NLRP3 activation. Formation of ASC oligomers reflects inflammasome activation. By applying an established cross-linking protocol followed by western blot it is possible to detect and discriminate between ASC monomers or oligomers using a specific anti-ASC antibody [52,118,119]. Another powerful approach to observe ASC oligomerization is the detection of ASC specks, which reflect massive activation of the NLRP3 inflammasome that precedes pyroptosis [22,116,120,121]. Traditional immunofluorescence staining techniques have been used to detect ASC specks using a specific anti-ASC antibody. As an alternative, human and murine cell lines have been developed that stably express a construct containing ASC-mCerulean, ASC-mCherry, or ASC fused with other fluorescent tags [122,123,124]. By using these cell lines it is possible to monitor the formation of ASC specks in live mode, with no need for staining [59]. Very recently, a new flow cytometry-based approach has been reported for the detection of ASC specks in activated human PBMCs [125]. Briefly, PBMCs are stained for ASC and CD14 (monocyte marker) and analyzed by flow cytometry. The analysis consists of gating ASC^+^ cells in the monocyte population and analyzing the distribution of ASC-FITC width vs ASC-FITC area in CD14^+^ monocytes in order to determine and quantify the percentage of ASC specking monocytes. This method can also be applied in other cell types including THP-1, J774A.1, and BMDMs [125]. Immunofluorescence can be used to study protein expression and localization. For example, it has been reported that NLRP3 localization on mitochondria membranes under certain circumstances is required for optimal inflammasome activation [91,102,126].

Co-immunoprecipitation (Co-IP) is a technique used to study protein–protein interactions. It can be performed on endogenous proteins or in recombinant systems, such as HEK293T, where the proteins of interest are co-transfected. Co-IP assays use antibodies specific to a target protein to indirectly capture proteins that are bound to the target one. Pull-down of antibody-bound proteins is normally performed using agarose or magnetic beads. Further downstream analysis, such as Western Blot, is usually used in order to check whether the protein of interest has been pulled down together with the target protein [127]. Furthermore, Co-IP experiments are used to assess the effect of a given molecule on protein-protein interaction. For example, Co-IP experiments in HEK293T cells have been performed to assess the impact of CY-09 and MCC950 on NLRP3-NLRP3 and NLRP3-ASC interaction [52,59].

### 3.2. Probe-Based Assays

Probe-based assays include different approaches that use colorimetric, fluorimetric, or luminescent probes to directly monitor a specific event in cell-based or cell-free assays. For instance, fluorescent carboxyfluorescein-labeled inhibitor of caspases (FAM-FLICA) probes are widely used to detect active caspases in cells using a fluorescence plate reader, flow-cytometry or fluorescence microscopy. Briefly, FAM-FLICA reagents are cell permeable and irreversibly react with active caspases inside the cell, releasing a fluorescent signal only when bound to caspases. Specificity is conferred by a tetra-peptide incorporated in the substrate that can be recognized by a given caspase. For example, the FLICA reagent FAM-YVAD-FMK specifically detects active caspase-1. Upon covalent reaction with active caspase-1, the fluorescent probe is retained within the cell, while any unbound FAM-YVAD-FMK diffuses out of the cell and it is washed away. The remaining green fluorescent signal is a direct measure of active caspase-1 present inside the cell [12,128].

Recently, a bioluminescence assay has been developed to monitor caspase enzymatic activity in cell-free extracts or cell supernatants [129]. As for the FAM-FLICA probes, specificity for a given caspase is conferred by the peptide sequence of the bioluminescent substrate. Assay specificity can be further enhanced by the addition of proper controls [12]. For example, to measure the activity of caspase-1, the substrate Z-WEHD-aminoluciferin, which incorporates the optimal caspase-1 recognition tetrapeptide, has been created. Active caspase-1 cleaves the substrate leading to aminoluciferin release, thus resulting in the luciferase reaction and light production that can be measured with a luminescence microplate reader [129].

### 3.3. Cell Death Assays

As above mentioned, massive activation of the NLRP3 inflammasome leads to pyroptosis, a mechanism of cell death different from the well-known apoptosis. Pyroptosis is triggered by the formation of GSDMD pores on the cell membrane and the consequent release of cytosolic proteins. Quantification of extracellular lactate dehydrogenase (LDH) release is a typical read-out commonly used for evaluating any form of cell death associated with rupture of cell membrane [116,130]. Several commercial kits are available that allow the quantification of extracellular LDH using luminescent, fluorescent, or colorimetric readout. The procedure is normally very quick and can be performed in two steps. Due to its simplicity and effectiveness, this assay has been widely used to test the efficacy of selective NLRP3 inhibitors [59,116,120,130].

### 3.4. Surface Plasmon Resonance (SPR)

Surface plasmon resonance (SPR) is often used to determine the dissociation constant (“binding constant”, *K*_D_) between a protein and its ligand. Normally, a bait ligand is immobilized on a sensor chip. Through a microfluidic system, a solution with the prey analyte is injected on the bait layer. From the association (“on rate,” *k*_a_) and dissociation rates (“off rate” *k*_d_) of the bait ligand and the prey analyte, it is possible to calculate the equilibrium dissociation constant (“binding constant” *K*_D_) as a ratio. By using this approach Hu and collaborators showed that, in Muckle–Wells Syndrome, the NLRP3-D31V mutation enhances the binding of NLRP3 with ASC resulting in an over-production of IL-1β and excessive immune responses including periodic fever, arthralgia and occasional conjunctivitis [131]. Using the same approach, Lee et al. demonstrated that caffeic acid phenethyl ester (CAPE) can bind directly ASC, resulting in blockade of NLRP3-ASC interaction induced by MSU crystals [132]. Finally, Coll et al. found, by SPR analysis, that MCC950 directly interacts with the Walker B motif within the NLRP3 NACHT domain, thereby blocking ATP hydrolysis and inhibiting NLRP3 activation and inflammasome formation [48].

**Table 2 ijms-21-04294-t002:** Assays and their applications in inflammasome biology.

What Can be Detected	Type of Assay	Sample Type	Quantitative (Y/N)	Readout	Refs
**Soluble proteins (i.e., IL-1** **β** **, IL-18)**	ELISA	Supernatants	Y	Absorbance	[14,116,117]
	TR-FRET-based assays	Supernatants	Y	Fluorescence	[112,114]
	AlphaLISA-based assays	Supernatants	Y	Luminescence	[113,115]
**Protein expression**	Western blot	Cell Lysates	N	Chemi/ Fluorescence	[14,52,118,119]
	(Immuno)fluorescence	Living/fixed cells or Tissue	N	Fluorescence	[22,59,102,116,120,121,122,123,124,125]
**Protein processing (i.e., pro- IL-1** **β** **, pro-caspase-1)**	Western blot	Supernatants	N	Chemi/ Fluorescence	[12,116,117]
	BRET-based probes	Living cells	Y	Bioluminescence	[69,70]
**ASC Oligomers**	Western blot	Cell Lysates	N	Chemi/ Fluorescence	[50,118,119]
**ASC Specks**	(Immuno)fluorescence	Living/fixed cells or Tissue	Y *	Fluorescence	[22,116,120,121]
	Flow cytometry	Living/fixed cells	Y	Fluorescence	[125]
**Active caspases**	Fluorescence (probe-based)	Living/fixed cells or Tissue	N	Fluorescence	[12]
	Flow cytometry	Living/fixed cells	Y	Fluorescence	[128]
**Caspase activity**	Enzymatic assay (probe-based)	Cell free extracts/supernatants/recombinant enzyme	Y	Luminescence	[12,129]
**Protein-protein Interaction**	SPR	Cell-free extract/Recombinant proteins	Y	Response/ Resonance Units	[48,131,132]
	Co-IP	Cell Lysates	N	Chemi/ Fluorescence	[52,59,127]
	BRET-based probes	Living cells	Y	Luminescence	[47,133]
**Conformational changes**	BRET-based probes	Living cells	Y	Luminescence	[133]
**Cell death**	LDH release	Supernatants	Y	Absorbance	[59,116,120,130]

* when ASC specks are expressed as % of total counted cells. Abbreviations: ELISA, enzyme-linked immunosorbent assay; TR-FRET, time-resolved fluorescence resonance energy transfer; IL-1β, Interleukin-1 beta; IL-18, Interleukin-18; BRET, bioluminescence resonance energy transfer; ASC, apoptosis-associated speck-like protein containing a CARD; SPR, surface plasmon resonance; Co-IP, co-immunoprecipitation; LDH, lactate dehydrogenase.

### 3.5. Bioluminescence Resonance Energy Transfer (BRET)-Based Assays

Bioluminescence resonance energy transfer (BRET) technology allows the user to monitor protein–protein interaction, protein cleavage, and conformational changes in living cells and in cell-free systems. In the context of inflammasome research, several examples of BRET-based assays have been reported. For example, a BRET-based approach was proposed for the study of the interaction between NLRP3 proteins in living cells [133]. Another example reported by Pelegrin et al. developed a BRET-based biosensor to monitor pro-IL-1β processing in living cells using a plate reader or a microscope. Specifically, in this example, pro- IL-1β was fused at its terminals with a donor (Rluc8) and an acceptor (Venus). The proximity of the two molecules leads to energy transfer for the detection of the protein. However, when pro-IL-1β is cleaved, donor and acceptor become distant and the BRET signal is reduced [69,70]. More recently, BRET-based approaches have been used to study the molecular conformation of NLRP3. For example, Hafner-Bratkovic et al. reported a construct were the donor (luciferase) and the acceptor (YFP fluorescent protein) probes were added intramolecularly (N-terminus and C-terminus of the same NLRP3 protein) to study NLRP3 change of conformation upon activation and intermolecularly (donor and acceptor probes in different NLRP3 proteins) to follow NLRP3 oligomerization upon different stimuli [40]. Similarly, Tapia-Abellan et al. used a similar approach to demonstrate that the MCC950 inhibits NLRP3 activation by closing the active conformation into an inactive one [47].

## 4. Concluding Remarks

Recent advances in inflammasome research have provided new information on host defense mechanisms and unveiled a key role for NLRP3 in the development of several chronic-inflammatory and age-related diseases. The growing interest on NLRP3 biology calls for a parallel increase in the development of methods that are necessary for molecular and biochemical studies. As herein reported, several established cellular models are available and used throughout the world to investigate inflammasome biology. However, they cannot always recapitulate the behavior of diseased macrophages in inflammatory NLRP3-mediated pathologies. For this reason, major efforts have to be made toward the establishment of patient-derived iPS-DM in vitro. The development of these models will speed up the process of defining new patient-specific therapies for NLRP3-related disease.

Furthermore, several established cellular, biochemical, and biophysical assays exist for the study of NLRP3 inflammasome biology and continuous efforts are done toward the development of novel, quantitative, and specific approaches. An area that will require special attention in the near future will be the study of NLRP3 from a biochemical and structural point of view. In fact, a high-resolution 3D structure of NLRP3 is still missing, and the production of the recombinant NLRP3 protein still poses technical challenges. Therefore, future research efforts should be directed toward filling these gaps. The knowledge that will be generated will surely represent a breakthrough for the study of NLRP3 biology and the development of selective drugs.

## Figures and Tables

**Figure 1 ijms-21-04294-f001:**
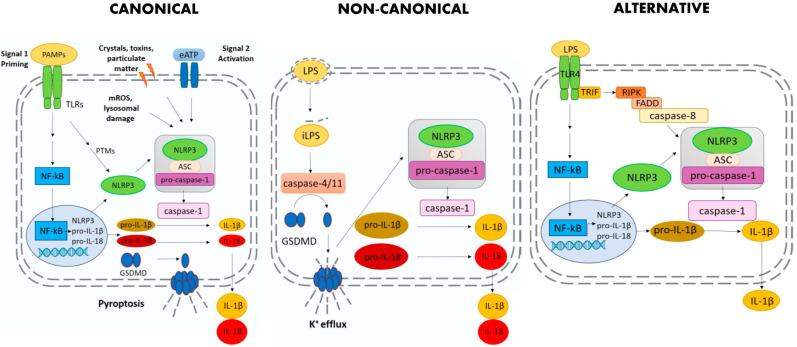
Schematic representation of the mechanisms regulating inflammasome activation in canonical, non-canonical and alternative pathway. Abbreviations: PAMP, pathogen-associate molecular pattern; TLRs, toll-like receptors; NF-kB, nuclear factor kappa-light-chain-enhancer of activated B cells; PTM, post-translational modifications; eATP, extracellular adenosine triphosphate; mROS, mitochondrial reactive oxygen species; ASC, apoptosis-associated speck-like protein containing a CARD; IL, interleukin; GSDMD, gasdermin D; iLPS, intracellular LPS; TRIF, TIR-domain-containing adapter-inducing interferon-β; RIPK, receptor-interacting serine/threonine-protein kinase 1; FADD, Fas-associated protein with death domain; NLRP3, NLR family pyrin domain containing 3.

**Table 1 ijms-21-04294-t001:** Cell models used in NLRP3 inflammasome research.

Cell Model	Description	Source	Example of Applications	References
**BMDMs**	Primary bone-marrow-derived macrophages (wt and KO)	Mouse	Canonical and non-canonical activation; identification of inhibitors	[1,14,15,18,19,22,24,26,54,61,62,63,64]
**iBMDM**	Immortalized primary bone-marrow derived macrophages (wt and KO)	Mouse	Canonical and non-canonical activation; identification of inhibitors	[1,18,21,22,40,49,52,65,66,67,68,69,70]
**Raw264.7**	Macrophage-like cell line	Mouse	Priming events studies	[61,71,72,73]
**J774A.1**	Monocyte/macrophage cell line	Mouse	Canonical activation; identification of inhibitors	[53,69,70,71,74,75,76,77,78,79]
**hMDMs**	Primary human monocyte-derived macrophages	Human	Canonical and non-canonical activation; identification of inhibitors	[11,12,22,24,48,52,54,80,81,82,83,84,85]
**THP-1**	Monocyte-like cell line, (from Acute Monocytic Leukemia)	Human	Canonical and non-canonical activation; identification of inhibitors	[22,37,42,52,86,87,88,89,90,91,92,93,94]
**U937**	Monocyte-like cell line (from pro-monocytic Myeloid Leukemia)	Human	Canonical activation; identification of inhibitors	[22,53,95]
**BlaER1**	Human monocytes/macrophages (from immortalized B cells)	Human	Alternative activation	[32,86,96,97,98]
**HEK293T**	Human embryonic cell line	Human	Mechanistic studies	[22,37,40,42,48,49,52,99,100,101,102,103]
**iPS-DM**	iPS-derived macrophages	Human	Human auto-inflammatory disease studies	[86,104,105,106,107,108,109,110,111]

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
