# Peer review of "Cellular Models and Assays to Study NLRP3 Inflammasome Biology"

_ijms, 2020, doi:10.3390/ijms21124294_

Round 1
Reviewer 1 Report
Zito et al. summarized and discussed cellular models and cellular, biochemical and biophysical assays that are currently available for studying inflammasome biology and applying for developing selective small molecular inhibitors of NLRP3 inflammasome. Overall, this topic is well chosen and surely of interest in the field. However, this article surely needs to be improved in contents, and I have several major concerns, as follows. I hope these comments help improve the quality of this article.
- Inflammasome activation is induced during the inflammatory responses. Therefore, it is recommended for the authors to briefly introduce ‘Inflammation’ in the first paragraph of the Introduction section.
- In 1.1. Mechanisms of NLRP3 activation section, description of non-canonical activation of NLRP3 inflammasome is too simple. Please add the description of caspase-11 non-canonical inflammasome, functional cross-talk between caspase-11 and NLRP3 inflammsomes, and the mechanism by which caspase-11 non-canonical inflammasome-induced NLRP3 inflammasome activation (please see below literature).
- Yi YS. Functional crosstalk between non-canonical caspase-11 and canonical NLRP3 inflammasomes during infection-mediated inflammation. Immunology. 2020 Feb;159(2):142-155.
- Figure 1 is small, and it is hard to recognize the letters. Please enlarge this figure.
- In 1.3. Inhibition of NLRP3 for the treatment of inflammatory diseases section, please add the studies investigating the pharmacological effect on inflammatory and autoimmune diseases by targeting NLRP3 inflammasome and cite corresponding literature.
- In table 1., the correct description of RAW264.7 cell is ‘macrophage-like cell line’. Please modify it.
- In Concluding remarks section, the authors only summarized the contents of this article Please discuss the limitations of current assays that the authors described in the article, and provide the potential better assay methods or perspectives to overcome these limitations.
- There are several typo errors in the manuscript. Please go over the entire manuscript carefully and correct all the typos and grammatical errors.
Author Response
Zito et al. summarized and discussed cellular models and cellular, biochemical and biophysical assays that are currently available for studying inflammasome biology and applying for developing selective small molecular inhibitors of NLRP3 inflammasome. Overall, this topic is well chosen and surely of interest in the field. However, this article surely needs to be improved in contents, and I have several major concerns, as follows. I hope these comments help improve the quality of this article.
- Inflammasome activation is induced during the inflammatory responses. Therefore, it is recommended for the authors to briefly introduce ‘Inflammation’ in the first paragraph of the Introduction section.
A: We understand the Reviewer’s point of view and agree. Following his/her suggestion, we have modified the introduction of the manuscript by adding a small paragraph briefly describing the concept of “Inflammation” (lines 31-37).
- In 1.1. Mechanisms of NLRP3 activation section, description of non-canonical activation of NLRP3 inflammasome is too simple. Please add the description of caspase-11 non-canonical inflammasome, functional cross-talk between caspase-11 and NLRP3 inflammsomes, and the mechanism by which caspase-11 non-canonical inflammasome-induced NLRP3 inflammasome activation (please see below literature).
Functional crosstalk between non-canonical caspase-11 and canonical NLRP3 inflammasomes during infection-mediated inflammation. Immunology. 2020 Feb;159(2):142-155.
A: We apologize if we simplified too much the mechanism of non canonical activation. According to the suggestion of the Reviewer, in the revised manuscript we have expanded the paragraph describing the mechanisms of non canonical activation and have cited the suggested literature (lines 114-124)
- Figure 1 is small, and it is hard to recognize the letters. Please enlarge this figure.
A: We understand the Reviewer’s concern and agree. Accordingly, we have enlarged the fonts of the figure and will upload the Figure 1 as a separate file so that it will be possible to reach a satisfying resolution.
- In 1.3. Inhibition of NLRP3 for the treatment of inflammatory diseases section, please add the studies investigating the pharmacological effect on inflammatory and autoimmune diseases by targeting NLRP3 inflammasome and cite corresponding literature.
We thank the Reviewer for this suggestion. Accordingly, in paragraph 1.3 we have added a brief description of the studies investigating the effects of selective NLRP3 inhibitors on inflammatory diseases and cited the corresponding literature (lines 246-249).
- In table 1., the correct description of RAW264.7 cell is ‘macrophage-like cell line’. Please modify it.
A: We edited the table according to the suggestion
- In Concluding remarks section, the authors only summarized the contents of this article Please discuss the limitations of current assays that the authors described in the article, and provide the potential better assay methods or perspectives to overcome these limitations.
A: We thank the Reviewer for the suggestion. We edited the concluding remarks paragraph accordingly (lines 630-633 and 654-659).
- There are several typo errors in the manuscript. Please go over the entire manuscript carefully and correct all the typos and grammatical errors.
A: We apologize for the typo errors. The manuscript has been thoroughly revised before the resubmission
Reviewer 2 Report
The review of Zito et al. is summarizing cellular models to investigate the mechanisms and biology of NLRP3 inflammasome. The topic of this manuscript is interesting. However, I have some remarks and suggestions.
- Line 16: Please define the abbreviation “IL”.
- Line 34: Please define “ASC”.
- Line 36-38: Please rephrase “triggering the cleavage and release of the pro-inflammatory cytokines pro-IL-1β and pro-IL-18”. The bio inactive pro-forms of IL-1β and IL-18 are not released!
- Line 45-46: Please give a reference for “downregulation of NLRP3 has minor impact on host defence mechanisms”. Furthermore “defence” should be defense.
- Line 88: Inconsistent style “Alternative inflammasome activation: In 2016, a new…” in comparison to the paragraphs before.
- Line 90: Please add a reference.
- Line 93: Please define “TRIF-RIPK1-FADD”.
- Line 108-110: “An 108
- intact and functional NACHT domain is required for interaction with ASC, activation of caspase-1 and IL-1 release in THP-1 cells” please add a reference.
- Line 134: Please add a reference.
- Line 169: Inconsistent style “Primary Murine Bone-Marrow-Derived Macrophages (BMDMs). BMDMs have been a very…”. Maybe put a colon. See also line 189, 212.
- Line 213: Abbreviation for PBMCs was given earlier in the text.
- Line 252: Please add a reference.
- Line 275: Please rephrase the sentence and make clear that not untreated HEK293 cells were used. HEK cells were used as an expression system.
- Line 336 and Table 2: please define “ELISA”.
- Line 345-348: “Western blot on supernatant precipitates is the preferred approach to evaluate the processing of pro-IL-1, pro-IL-18, GSDMD and caspases. In this respect, the choice of antibodies will determine which one among the pro-form and the cleaved forms can be observed.” How is it possible to detect pro-forms in the supernatants? Please rephrase.
- Line 383: Please define “FAM-FLICA”.
- Figure 1, Table 1, Table 2: Please define the abbreviated words in the figure legend.
Author Response
The review of Zito et al. is summarizing cellular models to investigate the mechanisms and biology of NLRP3
inflammasome. The topic of this manuscript is interesting. However, I have some remarks and suggestions.
Line 16: Please define the abbreviation “IL”.
A: we have addressed The Reviewer’s request and have defined the abbreviation IL the first time it has been
mentioned (Line 16)
Line 34: Please define “ASC”.
A: According to the request of the Reviewer we have defined the abbreviation ASC the first time it has been
mentioned (Line 42)
Line 36-38: Please rephrase “triggering the cleavage and release of the pro-inflammatory cytokines
pro-IL-1β and pro-IL-18”. The bio inactive pro-forms of IL-1β and IL-18 are not released!
A: We thank the Reviewer for this comment and corrected the text according to his/her suggestion as follow:
“triggering the cleavage and release of the pro-inflammatory cytokines IL-1β and IL-18” (line 50)
Line 45-46: Please give a reference for “downregulation of NLRP3 has minor impact on host defence
mechanisms”. Furthermore “defence” should be defense.
A: We added a reference related to Reviewer’s request (ref #10) and edited “defence” with “defense” (Line 59)
Line 88: Inconsistent style “Alternative inflammasome activation: In 2016, a new…” in comparison to
the paragraphs before.
A: We thank the Reviewer for the suggestion and modified the sentence in order to improve manuscript
consistency (Line 125).
Line 90: Please add a reference.
A: we added the reference as requested (ref #32) (Line 127)
Line 93: Please define “TRIF-RIPK1-FADD”.
A: We have addressed the request of the Reviewer request and have defined the abbreviation TRIF-RIPK1-
FADD the first time they have been mentioned (Line 129-131).
Line 108-110: “An intact and functional NACHT domain is required for interaction with ASC, activation
of caspase-1 and IL-1 release in THP-1 cells” please add a reference.
A: We added the reference as requested (ref #37) (Line 198)
Line 134: Please add a reference.
A: We added the reference as requested (ref #10) (Line 222)
Line 169: Inconsistent style “Primary Murine Bone-MarrowDerived Macrophages (BMDMs). BMDMs have been a very…”. Maybe put a colon. See also line
189, 212.
A: We understand the Reviewer’s concern. In order to improve text consistency, we have decided to slightly
modify the format of the chapters by adding subchapters (for example 2.1.1, 2.2.2 and so on).
Line 213: Abbreviation for PBMCs was given earlier in the text.
A: We removed the full extension of PBMCs as requested (Line 342).
Line 252: Please add a reference.
A: Reference 22 in line 394 is related to the mechanism of inflammasome activation.
Line 275: Please rephrase the sentence and make clear that not untreated HEK293 cells were used.
HEK cells were used as an expression system.
A: We thank the Reviewer for the suggestion and changed the sentence as follows “Among others, Song and
collaborators used transfected HEK293T cells to demonstrate that NLRP3 phosphorylation mediated by JNK1
is an essential priming event for inflammasome activation” (Line 417)
Line 336 and Table 2: please define “ELISA”.
A: We have addressed the Reviewer’s request and have defined the abbreviation ELISA the first time it has
been mentioned (Line 515).
Line 345-348: “Western blot on supernatant precipitates is the preferred approach to evaluate the
processing of pro-IL-1, pro-IL-18, GSDMD and caspases. In this respect, the choice of antibodies will
determine which one among the pro-form and the cleaved forms can be observed.” How is it possible
to detect pro-forms in the supernatants? Please rephrase.
A: We understand the point raised by the Reviewer. However, our statement refers to conditions where strong
activation of the NLRP3 inflammasome occurs. For example, when macrophages are stimulated using
established protocols where priming with LPS is followed by Nigericin or ATP, cell death occurs. As a
consequence, intracellular proteins, such as the pro-forms of interleukins and caspases, can be found in the
supernatants. As an example, herein we report western blot images from two publications (including one
publication from our group). Yellow areas highlight the data showing that procaspase-1, pro-IL-1β and pro-IL18 can be found in the supernatants under certain circumstances.
PICTURE IN THE PDF FILE ATTACHED MARKED AS FIGURE 4
From Cipolina et al (Sci Rep . 2016, ;6:37625. doi: 10.1038/srep37625): “Figure 4. 17-oxo-DHA, but not
FP, suppresses NLRP3 inflammasome activation in PBMCs. PBMCs from healthy individuals (N = 6) were
treated with 1 μ g/ml LPS for 3.5 h, followed by FP (10 nM) or 17-oxo-DHA (5 μ M) for 30 min, then by 10 μ M
nigericin for 30 min in serum-free medium. (a) IL-1β was measured in the supernatants by ELISA assay. Mean
with SEM are reported. *p-value < 0.05; ns, p-value > 0.05. (b) Precursor and cleaved caspase-1 and IL-1β
were measured by western blot in total cell extracts and supernatant precipitates, respectively. NLRP3 was
measured by western blot in total cell extracts. A western blot representative of three independent experiments
is shown.”
PICTURE IN THE PDF FILE ATTACHED AS FIGURE 1
From Palazòn-Riquelme et al (EMBO Rep . 2018, (10):e44766. doi: 10.15252/embr.201744766.) “Figure 1.
Inhibition of USP7 and USP47 with P22077 blocks human NLRP3 inflammasome activation. (A) IL‐1β ELISA
of supernatants from LPS‐primed (1 μg/ml, 4 h) MDMs pre‐incubated with 0.1% DMSO or P22077 (2.5 μM)
15 min before treatment with either nigericin (10 μM, 45 min) or CPPD crystals (250 μg/ml, 2 h). Bars represent
the mean ± SD, n = 13 and 11 independent blood donors for nigericin and CPPD, respectively. ***P < 0.001
using a one‐way ANOVA.
(B)IL‐18 ELISA of supernatants from MDMs treated as in (A). Bars represent the mean ± SD, n = 11
independent blood donors. ***P < 0.001, **P < 0.01 and *P < 0.05 using a one‐way ANOVA.
(C) LDH release from MDMs treated as in (A). Bars represent the mean percentage of LDH release relative to
the total cells lysed ± SD, n = 10 and 11 independent blood donors for nigericin and CPPD, respectively. ***P
< 0.001; **P < 0.01 using a one‐way ANOVA. n.s. = not significant.
(D) Western blots of supernatants and cell lysates from MDMs treated as in (A). Bands in the figure represent
the following: pro‐IL‐1β (31 kDa); mature IL‐1β (mIL‐1β, 17 kDa); pro‐IL‐18 (24 kDa); mature IL‐18 (mIL‐18,
18 kDa); pro‐caspase‐1 (pro‐Casp‐1, 45 kDa); and mature caspase‐1 (mCasp‐1, 20 kDa). β‐Actin is shown as
a loading control. Blots are representative of at least three independent blood donors.”
Line 383: Please define “FAM-FLICA”.
A: We have addressed the Reviewer’s request and have defined the abbreviation FAM-FLICA in the text (line
566)
Figure 1, Table 1, Table 2: Please define the abbreviated words in the figure legend.
A: For Figure 1 and Table 1 and 2 we have added, as requested, a legend including all the abbreviations.

Round 2
Reviewer 1 Report
The authors appropriately addressed all comments of the reviewer, and the article is now acceptable for publication.